Review Article

# The splice of life: how alternative splicing shapes regulatory and phenotypic evolution

Carissa Emerson Hunter[1,2,3] & Yi Xing [ID][1,4,5 ✉]

## Abstract

**Alternative splicing (AS) is a key mechanism for generating regulatory and phenotypic diversity in multicellular eukaryotes. Large-scale comparative transcriptomic studies have revealed that AS leads to lineage-specific and tissue-specific transcriptomic and proteomic changes, underscoring its contribution to the evolution of gene products and functions. In this review, we highlight the patterns and mechanisms of AS evolution across species, exploring how technological advancements are transforming our understanding of splicing evolution. Furthermore, we discuss mechanistic and functional insights from recent studies, including groundbreaking discoveries on how AS has shaped phenotypic evolution in mammals.**

**Keywords** Alternative Splicing; Exon Evolution; Splicing Regulation; Comparative Transcriptomics; Phenotypic Diversity
**Subject Categories** Evolution & Ecology; RNA Biology

## Introduction

It is well-established that both coding and non-coding changes in the genome play important roles in evolution through effects on protein function and gene regulation (King and Wilson, 1975). A prime example of regulatory evolution is the variation in beak morphology of Darwin's finches. Driven by relative BMP4 expression, beak size and shape mirror the selective advantage in different food sources (Abzhanov et al, 2004). While this work highlighted the critical role of gene expression in natural selection, changes in gene-level abundance capture only part of regulatory evolution. Moreover, a question persisted regarding the genetic underpinnings of organismal complexity. It was once thought that genome size was an indicator of organismal complexity, yet sequencing and analysis of the human genome revealed that its gene number is comparable to that of a worm (Elliott and Gregory, 2015; Ezkurdia et al, 2014). These observations have prompted a deeper exploration of regulatory evolution, with alternative splicing

(AS) emerging as an important layer beyond gene expression alone (Xing and Lee, 2006; Blencowe, 2006; Singh and Ahi, 2022).

Pre-mRNA splicing is a highly dynamic and coordinated RNA processing mechanism that removes introns and joins exons to produce mature mRNAs (Sharp, 1994). In many genes, alternative choices of exons and splice sites occur during splicing—a process known as AS that generates multiple distinct mRNA isoforms from a single gene (Nilsen and Graveley, 2010). Basic types of AS include exon skipping, alternative 5' or 3' splice sites, intron retention, and mutually exclusive exons (Fig. 1A). AS is mediated by an intricate program of trans-regulatory factors that bind to cis-regulatory elements within the pre-mRNA (Wang and Burge, 2008). Central to this process is the spliceosome, a large ribonucleoprotein (RNP) complex composed of multiple small nuclear RNP (snRNP) subunits, such as U1 and U2 snRNPs. To facilitate exon recognition, the spliceosome engages with core cis-elements, including the 5' and 3' splice sites, the branchpoint, and the polypyrimidine tract (Matera and Wang, 2014). Complementing this, auxiliary RNA-binding proteins (RBPs) bind to additional cis-regulatory elements, such as exonic and intronic splicing enhancers or silencers, to further modulate splicing outcomes (Fig. 1B).

Almost all multi-exon human genes undergo AS (Wang et al, 2008; Pan et al, 2008) and the different transcript isoforms generated can have distinct functional profiles, altering RNA stability, cellular localization, and protein-coding capacity (Kalsotra and Cooper, 2011). Many transcript isoforms are regulated in a tissue-specific or genetic manner, thereby influencing gene expression and diversifying protein production across cellular contexts and genetic backgrounds (Glinos et al, 2022; Park et al, 2018). This regulatory flexibility offers a versatile toolkit for evolutionary adaptation to occur.

## Patterns and mechanisms of AS evolution

Changes in AS over the course of evolution can reshape gene regulation and function, thereby contributing to phenotypic diversity. Major patterns of AS evolution include exon creation, exon loss, and shifts in exon splicing levels (Fig. 2). By comparing exon splicing across species, the ancestral state of an exon can be inferred to distinguish whether an exon was created, lost, or simply modified in its splicing activity (Xing and Lee, 2006; Alekseyenko

[1]Center for Computational and Genomic Medicine, The Children's Hospital of Philadelphia, Philadelphia, PA 19104, USA. [2]Genomics and Computational Biology Graduate Group, University of Pennsylvania, Philadelphia, PA 19104, USA. [3]Veterinary Medical Scientist Training Program, University of Pennsylvania School of Veterinary Medicine, Philadelphia, PA 19104, USA. [4]Department of Pathology and Laboratory Medicine, University of Pennsylvania, Philadelphia, PA 19104, USA. [5]Department of Biomedical and Health Informatics, The Children's Hospital of Philadelphia, Philadelphia, PA 19104, USA. ✉E-mail: xingyi@chop.edu

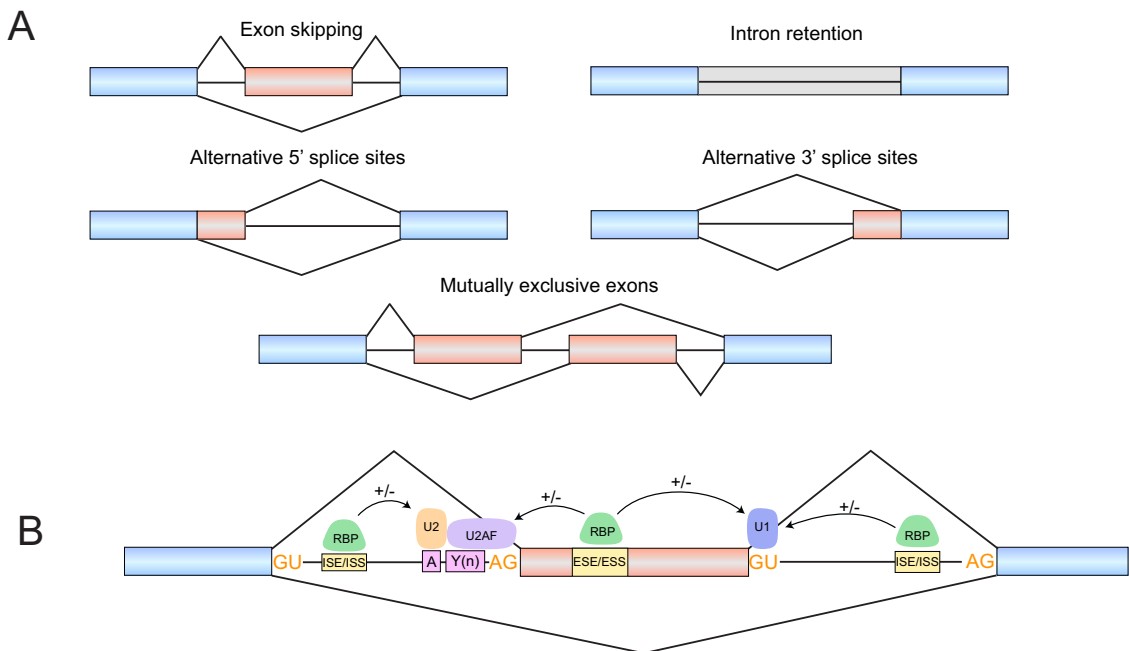

**Figure 1. Alternative splicing patterns and regulation.**

(A) Basic patterns of alternative splicing. Blue boxes represent constitutively spliced exons, and red boxes represent alternatively spliced exons. (B) Regulation of alternative splicing occurs through an intricate network of cis-acting elements and trans-acting factors. Cis-splicing signals include the 5′ and 3′ splice site motifs (GU/AG), the branchpoint ("A", pink box), and the polypyrimidine tract ("Y(n)", pink box), along with other cis-regulatory elements in exons (ESE: exonic splicing enhancer, ESS: exonic splicing silencer) and introns (ISE: intronic splicing enhancer, ISS: intronic splicing silencer). Trans-acting factors bind these elements, including U1 (binds 5′ splice site), U2 (binds branchpoint), and U2AF (U2 auxiliary factor, which binds the polypyrimidine tract and 3′ splice site), to regulate splicing.

et al, 2007). Exon creation frequently occurs through "exonization" in which intronic sequences, including those derived from transposable elements (TEs), are co-opted as new exons (Fig. 2A) (Sorek, 2007). Conversely, exon loss occurs when a constitutively spliced exon is eliminated through genomic deletion or weakened splice site recognition (Fig. 2B) (Wang et al, 2015). Finally, an exon may persist but display altered splicing activity due to cis-sequence changes that affect splice site strength or regulatory element recognition (Fig. 2C) (Lev-Maor et al, 2007). Other mechanisms, such as exon duplication or rearrangement, can also expand transcript diversity during evolution (Patthy, 1999; Martinez-Gomez et al, 2022). At the genetic level, most evolutionary changes in AS likely arise from cis-regulatory changes within the pre-mRNA. In an elegant study, Barbosa-Morais et al provided compelling evidence for this mechanism by analyzing a humanized mouse strain (Tc1) carrying human chromosome 21 (Barbosa-Morais et al, 2012). They found that the splicing patterns of human exons in this strain across multiple tissues closely resembled those of the identical exons in human tissues, rather than their orthologous mouse exons in native mouse tissues. These data suggest that lineage-specific changes in AS are predominantly driven by cis-sequence changes, rather than by changes in the concentration or activity of trans-acting factors, at least over the evolutionary distance between human and mouse (Barbosa-Morais et al, 2012).

Splicing outcomes are also shaped by trans-acting factors, including the core spliceosome machinery, splicing factors, or other RBPs, that interpret cis-splicing signals (Fu and Ares, 2014). In recent work, Rogalska et al systematically knocked-down 305 genes encoding splicing machinery components, revealing the extensive regulatory circuits through which these factors influence AS (Rogalska et al, 2024). Yet because most RBPs and core spliceosomal components are deeply conserved, much of the evolutionary changes in splicing still arise from alterations in their cis-binding landscapes. Lineage-specific sequence changes in splice sites or splicing enhancers and silencers reshape how these conserved factors interact with pre-mRNAs across species (Barbosa-Morais et al, 2012; Merkin et al, 2012). Across distantly related species, splicing factors with conserved spatiotemporal expression patterns and RNA-binding specificities may regulate vastly different sets of exons, due to evolutionary changes to cis-regulatory sequences of their target exons (Torres-Méndez et al, 2022; Márquez et al, 2021).

In some cases, trans-acting splicing factors themselves can acquire novel roles during evolution. For example, NOVA and its orthologues are critical neuronal splicing factors across metazoans and show divergent developmental expression patterns across species (Irimia et al, 2011; Gill et al, 2017; Ule et al, 2003). In sea urchin development, NOVA orthologues show expression confined to the postpharyngeal endoderm, whereas in Drosophila embryos they are absent from the central nervous system and restricted to salivary glands, gut, and fat bodies (Irimia et al, 2011; Weyn-Vanhentenryck et al, 2018). This theme is also seen with other splicing factors, like ESRPs, that have been recruited for different developmental roles across deuterostomes (Burguera et al, 2017). Lastly in rarer cases, molecular innovations within splicing factors

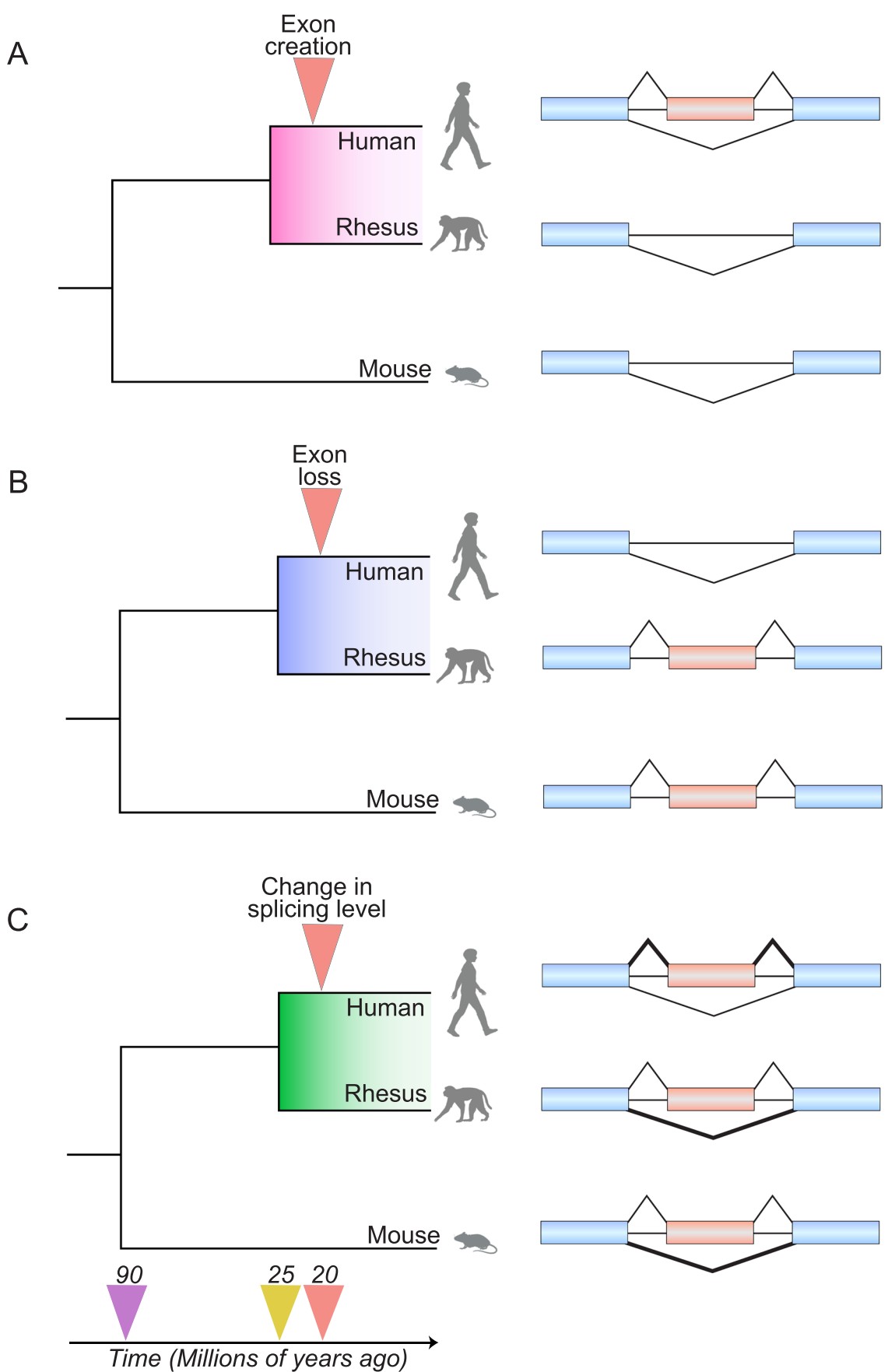

**Figure 2.   Different modes of splicing evolution.**

Phylogenetic trees showing mouse, rhesus, and human lineages over time (bottom arrow), with the mammalian lineage emerging ~90 million years ago (MYA, purple arrow) and the primate lineage emerging ~25 MYA (yellow arrow). Examples of different hominoid-specific splicing evolution are marked with a red arrow at ~20 MYA (red arrow). Blue boxes represent constitutive exons and red boxes represent the exon of interest with different evolutionary outcomes. (**A**) A hominoid-specific exon creation event, showing an exon that emerged after the divergence from monkey and mouse lineages. (**B**) A hominoid-specific exon loss event, showing an exon that was lost in humans, yet persists as a constitutive exon in mouse and monkey. (**C**) A hominoid-specific change in exon splicing level, showing increased inclusion in humans, but low inclusion in mouse and monkey. Splice junction line thickness indicates the relative frequency of exon inclusion or skipping. Animal images were created with BioRender (biorender.com).

themselves can also drive new splicing regulatory capabilities. A striking example involves the neuronal splicing factors SRRM3 and SRRM4. SRRM3 and SRRM4 are master regulators of microexons, a class of small exons (≤30 nucleotides) with highly conserved AS patterns necessary for neurodevelopment and function (Gonatopoulos-Pournatzis et al, 2020; Irimia et al, 2014; Ciampi et al, 2022). SRRM3 and SRRM4 acquired the enhancer of microexons (eMIC) domain early in bilaterian evolution. This evolutionary innovation enabled the spliceosome to specifically recognize and regulate microexons, thereby establishing conserved neurodevelopmental AS programs (Torres-Méndez et al, 2019). Similar cases of evolutionary changes in splicing factors that alter their functional activities or protein-protein interactions, thereby giving rise to lineage-specific splicing regulatory networks, have been reported for other splicing factors such as PTBP1 (Gueroussov et al, 2015) and members of the hnRNP A and D families (Gueroussov et al, 2017). Overall, evolution of cis-sequences and trans-factors creates a framework for the evolutionary tinkering of AS, opening new avenues for innovation and adaptation.

## Technological advances to study AS and isoform diversity

The ability to characterize AS and its evolution has been continuously advanced by technological revolutions in transcriptomics over the past two decades. While earlier studies of AS evolution based on expressed sequence tag (EST) sequencing (Modrek and Lee, 2003) and splicing-sensitive microarrays (Calarco et al, 2007; Lin et al, 2008) yielded important insights, the advent of high-throughput RNA sequencing (RNA-seq) has transformed the study of AS (Fig. 3) (Stark et al, 2019). For almost 15 years, short-read RNA-seq on the Illumina platform has been the predominant technology for transcriptomics, providing key insights into the prevalence of AS in multicellular eukaryotes (Wang et al, 2008; Pan et al, 2008; Tapial et al, 2017). However, short reads (50–200 bp) have inherent limitations for AS and isoform analysis, as they cannot easily resolve full-length transcript isoforms or accurately map repetitive and duplicated sequences in the transcriptome (Fig. 3A,B). The rise of long-read RNA-seq platforms, like those offered by Oxford Nanopore Technologies and Pacific Biosciences, have alleviated many of the limitations associated with short-read RNA-seq, especially when studying AS (Fig. 3C) (Pardo-Palacios et al, 2024). With reads spanning >10 kb in length, long-read RNA-seq can significantly improve the discovery and quantification of full-length transcript isoforms, particularly those generated from complex AS patterns (Amarasinghe et al, 2020; Monzó et al, 2025). While early long-read RNA-

seq studies were constrained by low throughput and high base error rate, continuous improvements in sequencing yield and accuracy have greatly expanded the capabilities of long-read RNA-seq, enabling accurate transcript discovery and quantification (Ament et al, 2025).

Transcriptomic studies often rely on bulk RNA-seq, but such data are obscured by the cellular heterogeneity present within a single tissue. Single-cell RNA-seq (scRNA-seq) has emerged as a powerful approach for transcriptome analysis, enabling the characterization of cell-type-specific transcriptomes at single-cell resolution (Birnbaum, 2018). With technological advances, scRNA-seq is no longer limited to capturing overall gene expression and can now characterize AS at the single-cell level. Short-read scRNA-seq approaches that are isoform-aware, such as Smart-seq3, can cover full-length transcripts (Hagemann-Jensen et al, 2020). As long-read RNA-seq has become increasingly popular, its integration into single-cell workflows has enabled isoform-resolved single-cell transcriptomics (Wen and Tang, 2026). For example, Joglekar et al created a comprehensive brain isoform atlas using long-read scRNA-seq of developing and adult human and mouse brains (Joglekar et al, 2024). Their work revealed cell-type and developmental regulation of transcript isoforms in genes linked to neuronal functions and diseases. While long-read scRNA-seq is powerful, a major bottleneck is the throughput of current long-read sequencing platforms (Ament et al, 2025). Recent technological advances, including programmed cDNA concatenation (Al'Khafaji et al, 2024) and targeted cDNA capture (Wang et al, 2023), may improve coverage and cost-efficiency.

With RNA-seq alone, the functional impact of transcript isoform variation often remains uncertain. The integration of RNA-seq data with proteomics data can link transcript isoforms with their corresponding protein isoforms, shedding light on the proteomic impact of AS and transcript isoform variation (Fig. 3D) (Nesvizhskii, 2014). Using ultra-deep proteomic analysis of human cells, Sinitcyn et al comprehensively profiled human variants and isoforms at the protein level (Sinitcyn et al, 2023). Their study suggested at scale that most frame-preserving AS events are indeed translated into proteins. Together, these technological advances in long-read, single-cell, and multi-omic analyses of AS enhance our ability to catalog transcript and protein isoforms and to elucidate their functional and evolutionary significance.

## Large-scale evolutionary studies of AS

Large-scale comparative transcriptomic studies have revealed widespread changes in AS during evolution. Two landmark studies

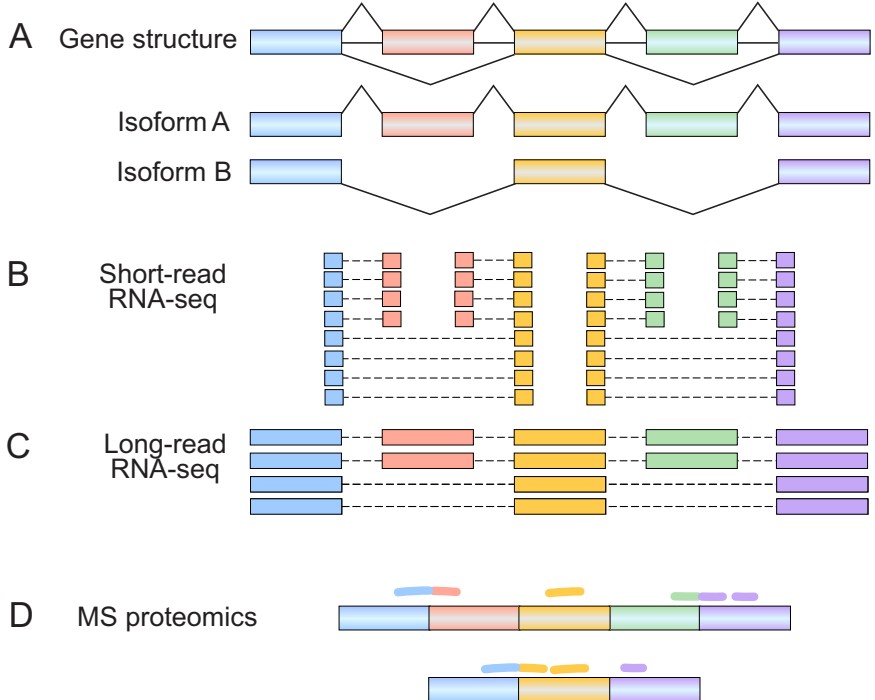

**Figure 3. Technological advances to study alternative splicing.**

(A) Schematic diagram of a gene that generates two alternatively spliced transcript isoforms (isoform A and isoform B). The first (blue), third (gold), and fifth (purple) boxes represent constitutive exons, while the second (red) and fourth (green) boxes represent two alternative exons that are co-spliced. Isoform A contains all five exons, while isoform B contains only the constitutive exons. (B) Short-read RNA-seq cannot easily resolve full-length transcript isoforms, failing to capture the long-range coupling of the alternative exons (red and green exons). (C) By contrast, long-read RNA-seq improves the discovery and quantification of full-length transcript isoforms, correctly capturing both transcript isoforms and linking distant exons (red and green exons) within isoform A. (D) Integration of mass spectrometry (MS)-based proteomics can link transcript isoforms with their corresponding protein isoforms, where peptide evidence (colored lines) can validate the translation of specific transcript isoforms.

in 2012 used RNA-seq to investigate the evolutionary landscape and dynamics of AS across vertebrates (Barbosa-Morais et al, 2012; Merkin et al, 2012). Barbosa-Morais et al compared tissue transcriptomes of vertebrate species spanning 350 million years of evolution (Barbosa-Morais et al, 2012). AS profiles diverged more rapidly as compared to gene expression profiles. When clustered based on gene expression profiles, samples clustered primarily by tissue type regardless of species. By contrast, when clustered based on AS profiles, samples clustered primarily by species, regardless of tissue type. Similar observations were made by Merkin et al (2012), highlighting the important role of AS in shaping lineage-specific tissue transcriptomes. Most lineage-specific AS events were driven by cis-regulatory changes, while a subset appeared to result from changes in trans-acting factors (Barbosa-Morais et al, 2012). Strikingly, splicing factors themselves can undergo lineage-specific AS events that drive downstream changes in the transcriptome. For instance, polypyrimidine tract binding protein 1 (PTBP1) is a well-characterized repressor of neural-specific AS events (Keppetipola et al, 2012). A mammalian-specific exon skipping event within *PTBP1* was shown to drive AS changes in downstream genes by weakening the functional activity of PTBP1, thereby contributing to mammalian neural development (Fig. 4) (Gueroussov et al, 2015). Collectively, these studies highlight extensive lineage- and tissue-specific differences in AS across and within species.

Building on these earlier findings, more recent work has carried out in-depth comparative analyses of AS and isoform variation within specific tissue types and developmental stages. Leung et al investigated the extent of transcript isoform diversity within the brain by comparing long-read RNA-seq data generated on human and mouse cerebral cortex samples (Leung et al, 2021). While most genes exhibited similar transcript isoform diversity between the human and mouse cortex (e.g., number of transcript isoforms), some genes showed striking interspecies differences. For example, *LPAR2* had 12 isoforms in the human cortex, yet only 1 isoform in the mouse cortex. Conversely, *Tmem191c* had 30 isoforms in the mouse cortex, yet only 1 isoform in the human cortex (Leung et al, 2021). Mazin et al investigated AS during organ development, comparing AS profiles across prenatal and postnatal stages in multiple organs from seven vertebrate species (Mazin et al, 2021). AS events that vary across developmental stages were substantially more conserved than those without developmental regulation. Furthermore, these AS events often preserved the reading frame, likely producing protein segments specific to developmental stage. Additionally, Mazin et al found that very young, species-specific exons showed higher inclusion in the testis, whereas older exons that originated in the eutherian ancestor showed higher inclusion in the brain (Mazin et al, 2021). This work supports the hypothesis that new exons emerge with initial expression in the transcriptionally permissive environment of the testis before co-option into the

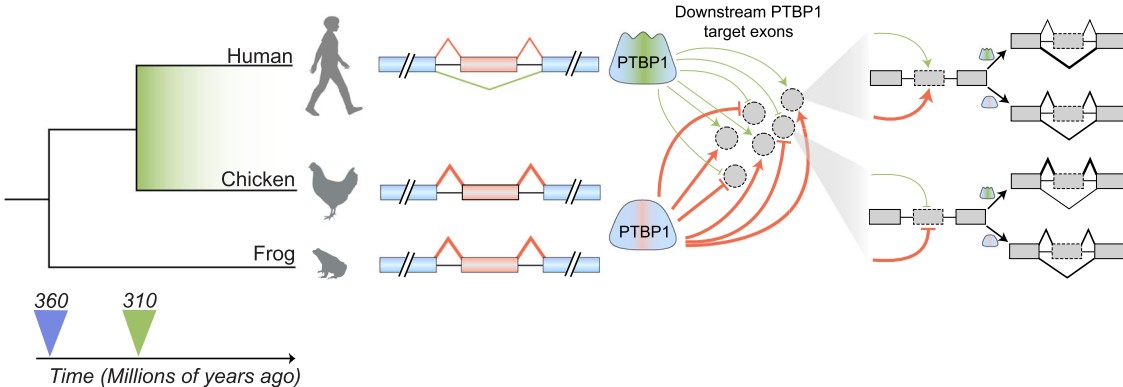

**Figure 4.  PTBP1 alternative splicing underlies lineage-specific splicing regulation.**

Phylogenetic trees showing human, chicken, and frog lineages over time (bottom arrow). The divergence of Amniota (mammals and birds) from Amphibia (frog) lineage occurred ~360 MYA (blue arrow), followed by a split within Amniota of mammals and birds ~310 MYA (green arrow). Polypyrimidine tract binding protein 1 (PTBP1) is a splicing factor that regulates the inclusion or skipping of numerous downstream target exons (gray circles). In mammals, the skipping of an exon (red box) within PTBP1 generates a protein isoform (green stripe) with weakened splicing regulatory activity (thin green lines). In contrast, the ancestral isoform (red stripe) constitutively includes this exon and exhibits stronger splicing regulatory activity (thick red lines). The different PTBP1 protein isoforms alter the splicing patterns of numerous target exons, as shown on the right: the PTBP1 skipping isoform leads to weakened exon inclusion (top) or weakened exon skipping (bottom) of PTBP1 target exons. Splice junction line thickness indicates the relative frequency of exon inclusion or skipping. Animal images were created with BioRender (biorender.com).

brain. Collectively, these data show that AS and isoform diversity vary substantially across species, tissues, and developmental stages, highlighting the interplay between developmental programs and evolutionary innovations in shaping transcriptomic and functional complexity.

An extensively studied mode of AS evolution is exon creation (Fig. 2A). Primate and human genomes contain thousands of newly created, lineage-specific exons (Sorek, 2007). Many primate-specific exons are derived from TEs, particularly the primate-specific Alu retrotransposon (Feschotte, 2008), though comparative RNA-seq analysis has also identified primate-specific exons created from non-TE intronic sequences (Merkin et al, 2015). Most newly created, primate-specific exons in the human genome have low splicing activities and are thought to represent evolutionary intermediates without established functions (Xing and Lee, 2006). Nonetheless, a subset have high splicing activities or tissue-specific splicing in normal human tissues (Lin et al, 2008; Shen et al, 2011). These exons are preferentially created and established in the 5'-UTR to regulate transcription (Merkin et al, 2015) or mRNA translation (Shen et al, 2011), and can also contribute novel peptides to alternative protein isoforms (Lin et al, 2016; Arribas et al, 2024). Collectively, these findings suggest that despite their very recent evolutionary origin, newly created exons can acquire strong splicing signals to significantly impact gene product and regulation.

A long-standing question in the field is whether AS complexity correlates with and contributes to organismal complexity (Nilsen and Graveley, 2010). Barbosa-Morais et al found substantial differences in AS complexity among vertebrate species, with the highest levels observed in primates (Barbosa-Morais et al, 2012). Increased AS complexity has also been observed over 1.4 billion years of eukaryotic evolution, with AS complexity strongly correlated with organismal complexity as measured by the number of cell types (Chen et al, 2014). While these observations might invite an adaptationist explanation, it has also been argued that the

high AS complexity observed in large-bodied animals can instead be explained by their small effective population sizes, suggesting that random genetic drift and relaxed purifying selection are major driving forces (Bénitière et al, 2024). The relative roles of adaptive evolution versus neutral drift in shaping AS evolution, both at the transcriptome scale and in specific genes, remain to be elucidated.

# Recent studies linking AS to phenotypic evolution

A major challenge in the study of AS lies in establishing the functional consequences of AS events. Although extensive AS differences have been observed between species, much less is known about the phenotypic impact of AS evolution. A long-standing framework posits that most AS changes over the course of evolution represent stochastic noise and neutral drift of the transcriptome (Gilbert, 1978; Bénitière et al, 2024). Newly created transcript isoforms are typically expressed at low levels, minimizing potentially deleterious effects and buffering interference with the original gene function. This relaxation of purifying selection provides an "evolutionary playground" in which lineage-specific AS events can persist and occasionally acquire new adaptive functions (Xing and Lee, 2006). Rather than inventing entirely new genes, AS tweaks existing genes to explore potential functions through new isoforms, without compromising the functions of existing ancestral isoforms. This principle parallels the role of gene duplication in generating novel functions (Force et al, 1999), consistent with the observed inverse correlation between AS and gene duplication, as well as the frequent loss of AS after gene duplication (Kopelman et al, 2005; Su et al, 2006).

While the aforementioned framework predicts that some lineage-specific AS events may have adaptive roles, concrete examples have been difficult to identify. Since 2024, however, a flurry of studies have linked AS events to phenotypic consequences

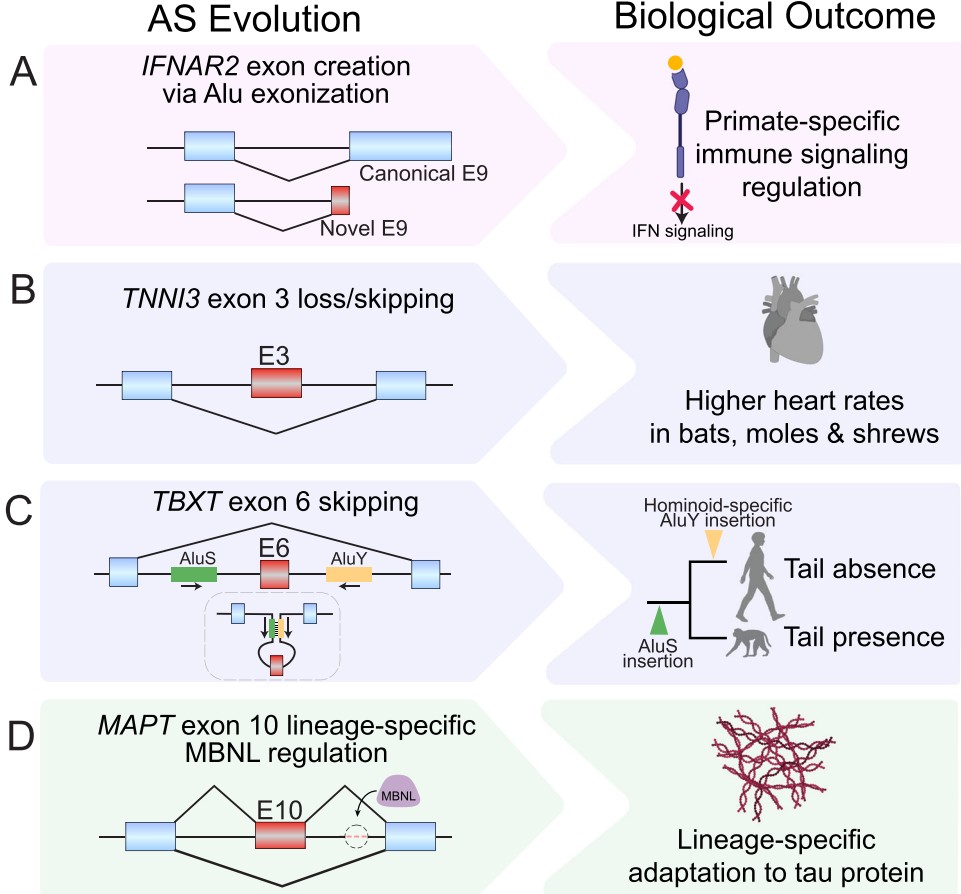

## AS Evolution

**A** — *IFNAR2* exon creation via Alu exonization
Canonical E9 / Novel E9

**B** — *TNNI3* exon 3 loss/skipping
E3

**C** — *TBXT* exon 6 skipping
AluS / E6 / AluY

**D** — *MAPT* exon 10 lineage-specific MBNL regulation
E10 / MBNL

## Biological Outcome

**A** — Primate-specific immune signaling regulation
IFN signaling

**B** — Higher heart rates in bats, moles & shrews

**C** — Hominoid-specific AluY insertion / Tail absence / AluS insertion / Tail presence

**D** — Lineage-specific adaptation to tau protein

**Figure 5. Case studies linking alternative splicing evolution to lineage-specific phenotypes.**

Each row illustrates an evolutionary alternative splicing (AS) change (left) and the resulting biological outcome (right). (**A**) An Alu exonization event within *IFNAR2* generates a novel terminal exon 9 (E9), creating a primate-specific truncated receptor isoform that lacks an intracellular signaling domain and modulates immune signaling in primates. (**B**) *TNNI3*'s exon 3 (E3) has been lost or inactivated across moles and shrews, and partially skipped via AS in bats, modifying cardiac troponin I calcium affinity and enabling heart rates that can reach up to 1500 beats per minute. (**C**) A hominoid-specific AluY insertion into the intron of a developmental gene, *TBXT*, can pair with an upstream inverted AluS to promote the skipping of exon 6 (E6), contributing to tail loss in hominoids. (**D**) Hominoid-specific weakening of a MBNL (purple) binding site (encircled pink dashed line) contributed to the lineage-specific shift in *MAPT*'s exon 10 (E10) splicing, shifting tau isoform proportion and driving lineage-specific change in tau protein. Schematics in right column were created with BioRender (biorender.com).

across diverse species and organ systems (Pasquesi et al, 2024; Plowman et al, 2025; Xia et al, 2024; Joyce et al, 2024; Recinos et al, 2024) (Fig. 5), demonstrating how AS evolution can directly shape lineage-specific traits. Here, we review examples of AS events that underpin lineage-specific adaptation and disease susceptibility.

Exon creation has occurred in thousands of human genes during primate and human evolution, primarily through the exonization of previously intronic sequences, particularly TEs. By analyzing long-read RNA-seq and proteomics datasets, Pasquesi et al comprehensively cataloged exonization events in primates, and identified the primate-specific Alu retrotransposon as a major source of new exons especially within immune genes (Pasquesi et al, 2024). They identified an Alu exonization event within *IFNAR2*, a gene encoding a subunit of the type I interferon receptor (Fig. 5A). The inclusion of this Alu exon generates a primate-specific truncated receptor isoform that lacks the intracellular signaling domain. The authors show that this new Alu-exon-containing isoform encodes a decoy receptor that dampens downstream JAK/

STAT signaling and transcription of interferon-stimulated genes. In another study, using short-read and long-read RNA-seq data, Plowman et al identified an anthropoid-specific isoform of *IL13RA1*, a gene encoding a subunit of the interleukin 13 receptor involved in the type 2 immune responses (Plowman et al, 2025). This isoform arose by TE exonization and produces a truncated receptor isoform that is partially defective in signaling capacity. In both studies, the novel isoform antagonizes the ancestral isoform, and the balance of these two isoforms was critical in maintaining immune homeostasis, highlighting the key role of exon creation in re-wiring immune signaling during evolution.

Complete exon loss or partial exon skipping can also shape phenotypic evolution. A classic example of complete exon loss is the human-specific deficiency in sialic acid N-glycolylneuraminic acid (Neu5Gc), caused by an Alu-mediated exon loss event within the CMP-N-acetylneuraminic acid hydroxylase (*CMAH*) gene (Chou et al, 2002). This exon loss event abolished *CMAH* gene activity and the production of Neu5Gc, and is thought to have

played a role in shaping pathogen susceptibility, reproductive biology, and possibly brain size during human evolution. Joyce et al showed another example of exon loss shaping phenotypic evolution, in this case affecting heart physiology (Fig. 5B) (Joyce et al, 2024). They focused on *TNNI3*, a gene encoding cardiac troponin I, a part of the troponin complex that plays an important role in cardiac contraction (Bers, 2002). Exon 3 of *TNNI3* encodes two serine residues (Joyce et al, 2024; Salhi et al, 2023) that when phosphorylated, decrease calcium affinity of the myofilaments (Wattanapermpool et al, 1995), increase the rate of cardiomyocyte relaxation (Yasuda et al, 2007), and facilitate diastolic filling (Marston and Pinto, 2022; Marston, 2023). The absence of exon 3 in *TNNI3* transcripts mimics the phosphorylated state and enables elevated heart rates. While humans exhibit nearly constitutive inclusion of exon 3, Joyce et al observed exon 3 to have been lost or inactivated across moles and shrews, and partially skipped via AS in bats, enabling heart rates that can reach up to 1500 beats per minute. This case illustrates that independent evolutionary events leading to either complete exon loss or partial exon skipping have occurred convergently across multiple species to drive similar physiological outcomes. Another recent striking finding of partial exon skipping contributing to phenotypic evolution was shown by Xia et al, who investigated the molecular underpinning of tail loss, a key phenotypic distinction between monkeys and apes (Fig. 5C) (Xia et al, 2024). The authors found a hominoid-specific AluY insertion into the intron of a developmental gene, *TBXT*, that could pair with an upstream inverted AluS to promote the skipping of exon 6. In mouse models, the relative abundances of *TBXT* isoforms, either full-length or skipping exon 6, dictated tail length morphologies. Mice only expressing the full-length isoform had normal tail lengths, while those with greater relative expression of the exon 6 skipping isoform had shortened or absent tails (Xia et al, 2024).

The above examples highlight that phenotypically relevant AS evolution often does not involve binary all-or-nothing changes in exon inclusion or skipping, but rather more subtle shifts in isoform proportion. By analyzing RNA-seq data from humans and six other primates and treating isoform proportion as a quantitative trait, Recinos et al identified 1170 exons with lineage-specific shifts in isoform proportion in the primate brain showing evidence of stabilizing selection (Recinos et al, 2024). They focused on *MAPT*, a gene encoding microtubule-associated protein tau. Tau isoform diversity is primarily determined by the inclusion or skipping of exons 2, 3, and 10: the inclusion of exon 2 together with the inclusion or skipping of exon 3 creates tau isoforms with 2N or 1N N-terminal inserts, and the inclusion or skipping of exon 10 creates tau isoforms with 4R or 3R microtubule-binding repeats (Andreadis, 2005; Goedert and Jakes, 1990). Recinos et al found that in hominoids, decreased inclusion of exon 10 correlated with increased inclusion of exon 2, fine-tuning tau isoform proportion. They further showed that exon 10 splicing in primates is developmentally regulated by the splicing factor MBNL. Moreover, the hominoid-specific weakening of MBNL-binding sites contributed to the lineage-specific shift in exon 10 splicing and tau isoform proportion (Fig. 5D) (Recinos et al, 2024). Importantly, 3R and 4R tau isoforms are normally expressed at about an equal molar ratio in the human brain (Andreadis, 2005) and shifts in this balance link this AS event to changes in microtubule binding affinity (Goedert and Jakes, 1990) and diseases involving tauopathies (Petry et al,

2023; Bowles et al, 2022; Fernández-Nogales et al, 2014). Despite these intriguing observations, whether this shift in tau isoform proportion contributed to hominoid-specific phenotypes remains to be determined.

## Concluding remarks

AS provides a flexible and powerful evolutionary strategy to create novel functions without requiring new genes. More than two decades of comparative transcriptomics have revealed widespread changes in AS during evolution. Recent studies are beginning to link AS changes in specific genes to concrete functional and phenotypic consequences. As such discoveries continue to accumulate, they will further establish AS as a key contributor to phenotypic evolution and lineage-specific traits.

Advances in this field have been tightly coupled with technological innovations in transcriptomics. New approaches such as long-read RNA-seq, single-cell transcriptomics, and multi-omic integration, now enable comprehensive catalogs of full-length transcript and protein isoforms across tissues and individual cell types, greatly enhancing the resolution and accuracy of AS and isoform analysis. Because AS is particularly prevalent in complex organ systems that may undergo adaptive evolution, such as the brain and immune system, cell-type-resolved analysis may reveal a broader set of evolutionary events and the molecular mechanisms that drive AS evolution in specific cellular contexts. With growing interest and expanding technological capacity for AS profiling and functional investigation, we can expect an increasing number of cases where AS evolution is directly linked to phenotypic outcomes, underscoring its role as a central layer of molecular innovation and adaptation.

## Peer review information

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

## Acknowledgements

This work was supported by a National Institutes of Health grant (R01HD114705).

## Author contributions

**Carissa Emerson Hunter**: Conceptualization; Visualization; Writing—original draft; Writing—review and editing. **Yi Xing**: Conceptualization; Supervision; Funding acquisition; Writing—original draft; Writing—review and editing.

## Disclosure and competing interests statement

The authors declare no competing interests.

