## [Peer Review File · The EMBO Journal]

The Splice of Life: How Alternative Splicing Shapes Regulatory and Phenotypic Evolution

Yi Xing and Carissa Hunter

Corresponding author: Yi Xing (xingyi@chop.edu)

Review Timeline:

Submission Date:	16th Sep 25
Editorial Decision:	7th Oct 25
Revision Received:	5th Nov 25
Editorial Decision:	24th Nov 25
Revision Received:	29th Nov 25
Editorial Decision:	4th Dec 25
Revision Received:	4th Dec 25
Accepted:	7th Dec 25

Editor: Yehu Moran

Transaction Report:

Dear Dr. Xing,

Thank you for submitting your manuscript for consideration by the EMBO Journal. It has now been seen by three expert referees whose comments are shown below. I believe you will find the majority of them highly constructive.

Given the referees' positive recommendations, I would like to invite you to submit a revised version of the manuscript, addressing the comments of all three reviewers. I especially encourage you to consider the comment by Referee #2 regarding the potential role of neutral forces (genetic drift) in the evolution of alternative splicing. This is certainly something that should be clearly mentioned and discussed in your review.

Please do not hesitate to contact me directly via email should you like to discuss any specific points regarding the revision. As we wish to fit your manuscript into the special issue on ecology and evolution we plan for late 2025 or early 2026, I would like to ask you to aim to submit your revision within one month. If you believe this is not possible, please let me know ASAP. I very much look forward to receiving your revision.

Yours sincerely,

Yehu Moran
Academic Editor
The EMBO Journal

We realize that it is difficult to revise to a specific deadline. In the interest of protecting the conceptual advance provided by the work, we recommend a revision within 3 months (5th Jan 2026). Please discuss the revision progress ahead of this time with the editor if you require more time to complete the revisions.

Referee #1:

In this review, Hunter and Xing discuss how alternative splicing (AS) evolves and contributes to phenotypic evolution. This is a very well written and timely piece, covering much of the relevant literature in a dynamic and entertaining manner. I have enjoyed reading it and I have only a few discretionary suggestions.

- 1) While I can see the value of introducing parts of the content of Section II in between I and III, I found that this sections substantially cuts the narrative flow of the review. Also, the relevance of the findings of long-read sequencing and proteomics for the study of AS evolution is still very limited. Therefore, I would suggest to move this section to the end, and word it as "future directions" and promises of new technologies, in line with the Conclusion section.
- 2) One concept I missed at the end of Section I is the fact that, despite many splicing factors are conserved across large phylogenetic distances (e.g. mammals to flies) in terms of biochemical function and even cell type-specific expression, the networks of exon targets they regulate are completely different (e.g., Marquez et al, Genome Biol 2021; Torres-Mendez et al, Sci Adv 2022).
- 3) When mentioning the lack of expression of Nova in Drosophila, please keep in mind these initial observations were done only in embryos. Subsequent work has shown that Nova is expressed in adult neurons where it plays important roles (e.g. Gill et al, Cell 2017).
- 4) When discussing the evolutionary origin of SRRM4 as the master regulator of microexons, I think it is more correct to mention both SRRM3 and SRRM4. Even though SRRM4 was first discovered and characterized, they play redundant roles and various studies have shown that SRRM3 is indeed more essential than SRRM4 in both human and zebrafish.
- 5) Page 5: "higher eukaryotes". I would avoid using this expression in an evolutionary context, as it is inaccurate. It is better to say multicellular eukaryotes or, even more precisely, plants and animals. Also, there are many studies that have provided key insights also into the prevalence of AS in unicellular eukaryotes, so it might not even be necessary to restrict the statement to "higher" eukaryotes. (On a related note: checking the two associated references [12,13], I realize they refer only to classic human studies, so perhaps these can also be enriched/improved).
- 6) It is unclear whether exon skipping of TBXT has driven the loss of the tail or it is a passenger of other evolutionary changes. If the authors wish to be more neutral about this, they could write "...partial exon skipping is associated with phenotypic evolution...".

Referee #2:

Alternative splicing (AS) is emerging as a major contributor to evolutionary innovations, adaptations, new phenotypes, and diseases. Hunter and Xing review large-scale transcriptomic studies to highlight cases and mechanisms by which AS contributes to evolution and phenotypes. The topic has recently been covered by several reviews (Verta & Jacobs TREE 2022, Singh & Ahi MEC 2022 [the only one cited in this manuscript], Wright & Jiggins Nat Rev Genet 2022 to name a few), but the authors succeed in highlighting aspects specific to mammals and humans in a novel way. The authors cover patterns and mechanisms of AS, technological advances in AS quantification, and links between AS and phenotypic evolution, disease, and lineage-specific traits. The review is very well written and a great pleasure to read. The authors highlight key points about AS evolution, but do not discuss the role of neutral evolution in creating AS diversity and patterns, which is important when discussing any evolutionary trends. The review gives many helpful examples of AS contributing to evolutionary novelty, lineage-specific phenotypes, and diseases. This reliance on examples somewhat obscures the larger picture, however, and I encourage the authors to balance the ample examples with more general interpretation and outlook on what the patterns observed mean for the role of AS in regulatory/phenotypic evolution, lineage-specific traits, why are some types of AS mechanisms more prevalent, etc. Overall, this review gives a good picture of AS and its role in evolution and with a better discussion of the contribution of neutral processes, as well as balancing examples with over-arching interpretations, I believe it will make a strong contribution to readers.

My specific comments below:

Section 1:

- It's not immediately clear what is meant with "evolution" here - mechanisms of both de-novo emergence of new exons and evolutionary changes in existing AS diversity due to natural selection are discussed. This section would benefit from a clearer distinction of these two topics.
- "...most evolutionary changes in AS can be traced back to cis-regulatory changes in the pre-mRNA" I'm not completely sure if this is indeed the case and no references are provided, so at the very least, this statement needs to be properly cited. It is not clear whether "evolutionary changes" here mean the emergence of new exons or changes in AS frequencies.
- Splicing evolution and RBPs. The short paragraph on RBP contribution is good, but it would be interesting to add a perspective on the contribution of RBP regulatory networks and interactions among RBPs to splicing evolution (in addition to protein-coding changes in RBPs as is portrayed now).

Section 3:

- The section takes a noticeably "adaptationist" perspective on AS and it would be beneficial to discuss the relative roles of adaptive versus neutral evolution of AS (drift), especially as the review concentrates on mammals and humans. There have been strong arguments recently supporting the "drift-barrier" hypothesis for increase in AS diversity in larger animals that typically have lower population sizes - supporting that the main driving force of high AS diversity in animals is not adaptation but genetic drift. See e.g. Bénitière, F., Necsulea, A. & Duret, L. Random genetic drift sets an upper limit on mRNA splicing accuracy in metazoans. *eLife* 13, RP93629 (2024).
- Please define "developmentally regulated AS events".

Referee #3:

This review highlights the central role of alternative splicing (AS) in generating transcriptomic and proteomic diversity and shaping evolutionary adaptation. The authors trace the patterns and mechanisms of AS evolution, emphasizing exon creation, loss, and changes in inclusion levels driven primarily by cis-regulatory sequence changes, with occasional innovations in trans-acting splicing factors. They illustrate how molecular innovations, such as the SRRM4 eMIC domain enabling microexon regulation, established new regulatory programs critical for neurodevelopment.

The review concludes that AS is a powerful evolutionary strategy for innovation, providing functional flexibility without requiring new genes. As sequencing and functional tools advance, the field is poised to uncover many more direct links between AS evolution and phenotypic outcomes, particularly in complex organ systems such as the brain and immune system.

The reviewer has a few comments on the arrangement of the article

1. Expand section I with more in-depth discussion of the exon gain/loss mechanism, including exon shuffling, and exon evolution driven by gene duplication, and provide examples to help readers envision the hypothesis. The current version is thin. The current Figure 2 does not add much information beyond Figure 4. The idea of trans-acting factor evolution should also be incorporated into Figure 2.
2. Section II is distant from the main theme of the review. The reviewer suggests that it be condensed and combined with Section IV. Figure 3 is not relevant and should be removed without loss of clarity.

Minor comments:

1. The reference should be cited before punctuation marks. An EMBO Journal style should be used.
2. On page 8, explain and provide citations for "Evolutionarily younger exons showed higher inclusion in the testis, whereas older exons showed higher inclusion in the brain, supporting the hypothesis that new exons emerged with initial expression in the testis before co-option into the brain."
3. Figure 2 suggests the exon evolution events, marked by red triangles, evolved 5 MYA and are human-specific. However, the cited references generally describe them as hominoid-specific (except ref 19). Unless additional evidence is provided, these events could date back ~20 MYA. The figure should be revised to avoid overstating the specificity.
4. Figure 4D: The example of tau isoform regulation in primates (MAPT exon 10) is presented as linked to human phenotype, but the mechanistic connection to a human-specific trait is not fully developed. The authors should clarify how isoform proportion shifts translate to human- or hominoid-specific phenotypes, beyond association with disease states..

This is a valuable and timely review that synthesizes a large body of literature, including recent discoveries linking AS to phenotypic evolution. With revisions to strengthen Section I with exon evolution mechanisms, streamline Section II, and clarify figures and examples, the manuscript will provide a stronger conceptual framework and greater impact for readers.

Detailed Responses to Reviewer Comments

We thank all three reviewers for their thoughtful evaluation of our work. Overall, the reviewers had a highly favorable assessment of our manuscript. Reviewer #1 described our manuscript as “well written”, “entertaining,” and said they “enjoyed reading it.” Reviewer #2 said it was “a great pleasure to read.” Reviewer #3 wrote that our manuscript is “timely” and “valuable.” Here, we address each of their comments and we believe that this has strengthened the overall quality of the manuscript. All new and revised text is highlighted in red in the revised manuscript.

Referee #1:

In this review, Hunter and Xing discuss how alternative splicing (AS) evolves and contributes to phenotypic evolution. This is a very well written and timely piece, covering much of the relevant literature in a dynamic and entertaining manner. I have enjoyed reading it and I have only a few discretionary suggestions.

We thank the reviewer for this overall positive summary of our manuscript and we tried to address all of the reviewer’s comments.

1) While I can see the value of introducing parts of the content of Section II in between I and III, I found that this sections substantially cuts the narrative flow of the review. Also, the relevance of the findings of long-read sequencing and proteomics for the study of AS evolution is still very limited. Therefore, I would suggest to move this section to the end, and word it as "future directions" and promises of new technologies, in line with the Conclusion section.

The reviewer’s comment prompted us to clarify the logical order of the manuscript. In section I, we introduce AS and its role in evolution. We then progress to explain the technologies used to study AS in section II. Then, we discuss large-scale studies of AS evolution in section III and then dissect case examples of the specific phenotypic impacts of AS evolution in section IV. It is important to note that the technologies described in section II -- including long-read RNA-seq and proteomics -- were essential in some of the landmark studies highlighted in section III and IV. With this in mind, we would like to keep the review structured in this way. To address the reviewer’s concern, we deleted some non-essential text in section II on a single-cell RNA-seq study to sharpen the focus of the content and refine the flow of the review between sections. Additionally, we incorporated more technological context in the description of case studies in section IV.

2) One concept I missed at the end of Section I is the fact that, despite many splicing factors are conserved across large phylogenetic distances (e.g. mammals to flies) in terms of biochemical function and even cell type-specific expression, the networks of exon targets they regulate are completely different (e.g., Marquez et al, Genome Biol 2021; Torres-Mendez et al, Sci Adv 2022).

We thank the reviewer for bringing up this point. To address this, we added this concept and associated references into our discussion of RBPs in section I:

“Across distantly related species, splicing factors with conserved spatiotemporal expression patterns and RNA binding specificities may regulate vastly different sets of exons, due to evolutionary changes to cis-regulatory sequences of their target exons (Torres-Méndez *et al*, 2022; Márquez *et al*, 2021).”

3) When mentioning the lack of expression of Nova in *Drosophila*, please keep in mind these initial observations were done only in embryos. Subsequent work has shown that Nova is expressed in adult neurons where it plays important roles (e.g. Gill *et al*, Cell 2017).

We thank the reviewer for bringing up this point. We have modified section I to add the Gill *et al*. reference and make the description of NOVA expression patterns clearer:

“For example, NOVA and its orthologues are critical neuronal splicing factors across metazoans and show divergent developmental expression patterns across species (Irimia *et al*, 2011; Gill *et al*, 2017; Ule *et al*, 2003). In sea urchin development, NOVA orthologues show expression confined to the postpharyngeal endoderm, whereas in *Drosophila* embryos they are absent from the central nervous system and restricted to salivary glands, gut, and fat bodies (Irimia *et al*, 2011; Weyn-Vanhentenryck *et al*, 2018).”

4) When discussing the evolutionary origin of SRRM4 as the master regulator of microexons, I think it is more correct to mention both SRRM3 and SRRM4. Even though SRRM4 was first discovered and characterized, they play redundant roles and various studies have shown that SRRM3 is indeed more essential than SRRM4 in both human and zebrafish.

The reviewer’s concern is well-taken. Per the reviewer’s suggestion, we included SRRM3 in our discussion of the evolution of the eMIC domain and added relevant references. The text now reads:

“A striking example are the neuronal splicing factors SRRM3 and SRRM4. SRRM3 & 4 are master regulators of microexons, a class of small exons (≤ 30 nucleotides) with highly conserved AS patterns necessary for neurodevelopment and function (Gonatopoulos-Pournatzis *et al*, 2020; Irimia *et al*, 2014; Ciampi *et al*, 2022). SRRM3 & 4 acquired the enhancer of microexons (eMIC) domain early in bilaterian evolution. This evolutionary innovation enabled the spliceosome to specifically recognize and regulate microexons, thereby establishing conserved neurodevelopmental AS programs (Torres-Méndez *et al*, 2019).”

5) Page 5: "higher eukaryotes". I would avoid using this expression in an evolutionary context, as it is inaccurate. It is better to say multicellular eukaryotes or, even more precisely, plants and animals. Also, there are many studies that have provided key insights also into the prevalence of AS in unicellular eukaryotes, so it might not even be necessary to restrict the statement to "higher" eukaryotes. (On a related note: checking the two associated references [12,13], I realize they refer only to classic human studies, so perhaps these can also be enriched/improved).

We appreciate the reviewer's thoughtful comment and we have changed the sentence to use "multicellular eukaryotes" instead. As suggested, we now also incorporated a reference that extends beyond classic human studies. The text now reads:

"For almost 15 years, short-read RNA-seq on the Illumina platform has been the predominant technology for transcriptomics, providing key insights into the prevalence of AS in multicellular eukaryotes (Wang *et al*, 2008; Pan *et al*, 2008; Tapial *et al*, 2017)."

6) It is unclear whether exon skipping of TBXT has driven the loss of the tail or it is a passenger of other evolutionary changes. If the authors wish to be more neutral about this, they could write "...partial exon skipping is associated with phenotypic evolution...".

We have revised the text from "...partial exon skipping **driving** phenotypic evolution..." to "...partial exon skipping **contributing to** phenotypic evolution...". This matches the language used in the original paper.

Referee #2:

Alternative splicing (AS) is emerging as a major contributor to evolutionary innovations, adaptations, new phenotypes, and diseases. Hunter and Xing review large-scale transcriptomic studies to highlight cases and mechanisms by which AS contributes to evolution and phenotypes. The topic has recently been covered by several reviews (Verta & Jacobs TREE 2022, Singh & Ahi MEC 2022 [the only one cited in this manuscript], Wright & Jiggins Nat Rev Genet 2022 to name a few), but the authors succeed in highlighting aspects specific to mammals and humans in a novel way. The authors cover patterns and mechanisms of AS, technological advances in AS quantification, and links between AS and phenotypic evolution, disease, and lineage-specific traits. The review is very well written and a great pleasure to read. The authors highlight key points about AS evolution, but do not discuss the role of neutral evolution in creating AS diversity and patterns, which is important when discussing any evolutionary trends. The review gives many helpful examples of AS contributing to evolutionary novelty, lineage-specific phenotypes, and diseases. This reliance on examples somewhat obscures the larger picture, however, and I encourage the authors to balance the ample examples with more general interpretation and outlook on what the patterns observed mean for the role of AS in regulatory/phenotypic evolution, lineage-specific traits, why are some types of AS mechanisms more prevalent, etc. Overall, this review gives a good picture of AS and its role in evolution and with a better discussion of the contribution of neutral processes, as well as balancing examples with over-arching interpretations, I believe it will make a strong contribution to readers.

We thank the reviewer for their enthusiastic assessment of our work.

My specific comments below:

Section 1:

- It's not immediately clear what is meant with "evolution" here - mechanisms of both de-novo emergence of new exons and evolutionary changes in existing AS diversity due to natural selection are discussed. This section would benefit from a clearer distinction of these two topics.

This reviewer feedback prompted us to modify the first few sentences of section I such that the definition is now immediately following the introductory sentence. Specifically, we consider exon creation, exon loss, and shifts in exon splicing levels (Figure 2) as the principal modes of AS evolution. The introduction of section I now reads:

“Changes in AS over the course of evolution can reshape gene regulation and function, thereby contributing to phenotypic diversity. Major patterns of AS evolution include exon creation, exon loss, and shifts in exon splicing levels (Figure 2). By comparing exon splicing across species, the ancestral state of an exon can be inferred to distinguish whether an exon was created, lost, or simply modified in its splicing activity (Xing & Lee, 2006; Alekseyenko *et al*, 2007).”

- "...most evolutionary changes in AS can be traced back to cis-regulatory changes in the pre-mRNA" I'm not completely sure if this is indeed the case and no references are provided, so at the very least, this statement needs to be properly cited. It is not clear whether "evolutionary changes" here mean the emergence of new exons or changes in AS frequencies.

We thank the reviewer for their comment. We have added the appropriate explanation and citation to support the above statement:

“At the genetic level, most evolutionary changes in AS likely arise from cis-regulatory changes within the pre-mRNA. In an elegant study, Barbosa-Morais *et al*. provided compelling evidence for this mechanism by analyzing a humanized mouse strain (Tc1) carrying human chromosome 21 (Barbosa-Morais *et al*, 2012). They found that the splicing patterns of human exons in this strain across multiple tissues closely resembled those of the identical exons in human tissues, rather than their orthologous mouse exons in native mouse tissues. These data suggest that lineage-specific changes in AS are predominantly driven by cis-sequence changes, at least over the evolutionary distance between human and mouse (Barbosa-Morais *et al*, 2012).”

- Splicing evolution and RBPs. The short paragraph on RBP contribution is good, but it would be interesting to add a perspective on the contribution of RBP regulatory networks and interactions among RBPs to splicing evolution (in addition to protein-coding changes in RBPs as is portrayed now).

The reviewer brings up an important point. To address this, we have added the following sentence to the discussion on RBP regulatory networks and interactions:

“Similar cases of evolutionary changes in splicing factors that alter their functional activities or protein-protein interactions, thereby giving rise to lineage-specific splicing regulatory networks, have been reported for other splicing

factors such as PTBP1 (Gueroussov *et al*, 2015) and members of the hnRNP A and D families (Gueroussov *et al*, 2017).”

Section 3:

- The section takes a noticeably "adaptationist" perspective on AS and it would be beneficial to discuss the relative roles of adaptive versus neutral evolution of AS (drift), especially as the review concentrates on mammals and humans. There have been strong arguments recently supporting the "drift-barrier" hypothesis for increase in AS diversity in larger animals that typically have lower population sizes - supporting that the main driving force of high AS diversity in animals is not adaptation but genetic drift. See e.g. Bénitière, F., Necsulea, A. & Duret, L. Random genetic drift sets an upper limit on mRNA splicing accuracy in metazoans. *eLife* 13, RP93629 (2024).

We thank the reviewer for bringing up this important point. To address this, we expanded the introduction of section IV to describe the conceptual framework of AS evolution as well as the relative roles of adaptive versus neutral evolution:

“A long-standing framework posits that most AS changes over the course of evolution represent stochastic noise and neutral drift of the transcriptome (Gilbert, 1978; Bénitière *et al*, 2024). Newly created transcript isoforms are typically expressed at low levels, minimizing potentially deleterious effects and buffering interference with the original gene function. This relaxation of purifying selection provides an “evolutionary playground” in which lineage-specific AS events can persist and occasionally acquire new adaptive functions (Xing & Lee, 2006). Rather than inventing entirely new genes, AS tweaks existing genes to explore potential functions through new isoforms, without compromising the functions of existing ancestral isoforms. This principle parallels the role of gene duplication in generating novel functions, consistent with the observed inverse correlation between AS and gene duplication, as well as the frequent loss of AS after gene duplication (Kopelman *et al*, 2005; Su *et al*, 2006).”

- Please define "developmentally regulated AS events".

We modified the text to clarify the above phrase. It now reads:

“AS events that vary across developmental stages were substantially more conserved than those without developmental regulation.”

Referee #3:

This review highlights the central role of alternative splicing (AS) in generating transcriptomic and proteomic diversity and shaping evolutionary adaptation. The authors trace the patterns and mechanisms of AS evolution, emphasizing exon creation, loss, and changes in inclusion levels driven primarily by cis-regulatory sequence changes, with occasional innovations in trans-acting splicing factors. They illustrate how molecular innovations, such as the SRRM4 eMIC domain enabling microexon regulation, established new regulatory programs critical for neurodevelopment.

The review concludes that AS is a powerful evolutionary strategy for innovation, providing functional flexibility without requiring new genes. As sequencing and functional tools advance, the field is poised to uncover many more direct links between AS evolution and phenotypic outcomes, particularly in complex organ systems such as the brain and immune system.

We thank the reviewer for their thoughtful assessment of our manuscript.

The reviewer has a few comments on the arrangement of the article

1. Expand section I with more in-depth discussion of the exon gain/loss mechanism, including exon shuffling, and exon evolution driven by gene duplication, and provide examples to help readers envision the hypothesis. The current version is thin. The current Figure 2 does not add much information beyond Figure 4. The idea of trans-acting factor evolution should also be incorporated into Figure 2.

We thank the reviewer for bringing up this point. We had previously included references to exon duplication and rearrangement in our original manuscript, but did not elaborate on these topics because recent literature is limited. Regarding gene duplication, we have now added the following text to section IV where we describe the conceptual framework of AS evolution:

“A long-standing framework posits that most AS changes over the course of evolution represent stochastic noise and neutral drift of the transcriptome (Gilbert, 1978; Bénitière *et al*, 2024). Newly created transcript isoforms are typically expressed at low levels, minimizing potentially deleterious effects and buffering interference with the original gene function. This relaxation of purifying selection provides an “evolutionary playground” in which lineage-specific AS events can persist and occasionally acquire new adaptive functions (Xing & Lee, 2006). Rather than inventing entirely new genes, AS tweaks existing genes to explore potential functions through new isoforms, without compromising the functions of existing ancestral isoforms. This principle parallels the role of gene duplication in generating novel functions, consistent with the observed inverse correlation between AS and gene duplication, as well as the frequent loss of AS after gene duplication (Kopelman *et al*, 2005; Su *et al*, 2006).”

Additionally, we have expanded section I to more deeply discuss trans-acting factor evolution. The text now reads:

“In some cases, trans-acting splicing factors themselves can acquire novel roles during evolution. For example, NOVA and its orthologues are critical neuronal splicing factors across metazoans and show divergent developmental expression patterns across species (Irimia *et al*, 2011; Gill *et al*, 2017; Ule *et al*, 2003). In sea urchin development, NOVA orthologues show expression confined to the postpharyngeal endoderm, whereas in *Drosophila* embryos they are absent from the central nervous system and restricted to salivary glands, gut, and fat bodies (Irimia *et al*, 2011; Weyn-Vanhentenryck *et al*, 2018). This theme is also seen with

other splicing factors, like ESRPs, that have been recruited for different developmental roles across deuterostomes (Burguera *et al*, 2017). Lastly in rarer cases, molecular innovations within splicing factors themselves can also drive new splicing regulatory capabilities. A striking example are the neuronal splicing factors SRRM3 and SRRM4. SRRM3 & 4 are master regulators of microexons, a class of small exons (≤ 30 nucleotides) with highly conserved AS patterns necessary for neurodevelopment and function (Gonatopoulos-Pournatzis *et al*, 2020; Irimia *et al*, 2014; Ciampi *et al*, 2022). SRRM3 & 4 acquired the enhancer of microexons (eMIC) domain early in bilaterian evolution. This evolutionary innovation enabled the spliceosome to specifically recognize and regulate microexons, thereby establishing conserved neurodevelopmental AS programs (Torres-Méndez *et al*, 2019). Similar cases of evolutionary changes in splicing factors that alter their functional activities or protein-protein interactions, thereby giving rise to lineage-specific splicing regulatory networks, have been reported for other splicing factors such as PTBP1 (Gueroussov *et al*, 2015) and members of the hnRNP A and D families (Gueroussov *et al*, 2017). Overall, evolution of cis-sequences and trans-factors creates a framework for the evolutionary tinkering of AS, opening new avenues for innovation and adaptation.”

Additionally, we have created a new figure (Figure 4, “PTBP1 alternative splicing underlies lineage-specific splicing regulation”) to illustrate how trans-factor evolution can give rise to lineage-specific splicing regulatory networks.

2. Section II is distant from the main theme of the review. The reviewer suggests that it be condensed and combined with Section IV. Figure 3 is not relevant and should be removed without loss of clarity.

The reviewer’s comment prompted us to clarify the logical order of the manuscript. In section I, we introduce AS and its role in evolution. We then progress to explain the technologies used to study AS in section II. Then, we discuss large-scale studies of AS evolution in section III and then dissect case examples of the specific phenotypic impacts of AS evolution in section IV. It is important to note that the technologies described in section II -- including long-read RNA-seq and proteomics -- were essential in some of the landmark studies highlighted in section III and IV. With this in mind, we would like to keep the review structured in this way. To address the reviewer’s concern, we deleted some non-essential text in section II on a single-cell RNA-seq study to sharpen the focus of the content and refine the flow of the review between sections. Additionally, we incorporated more technological context in the description of case studies in section IV.

Minor comments:

1. The reference should be cited before punctuation marks. An EMBO Journal style should be used.

We thank the reviewer for pointing this out. We have now ensured that all references are placed before punctuation marks and that the reference list is formatted according to The EMBO Journal style.

2. On page 8, explain and provide citations for "Evolutionarily younger exons showed higher inclusion in the testis, whereas older exons showed higher inclusion in the brain, supporting the hypothesis that new exons emerged with initial expression in the testis before co-option into the brain."

We have revised the text to clarify this point. The text now reads:

“Additionally, Mazin et al. found that very young, species-specific exons showed higher inclusion in the testis, whereas older exons that originated in the eutherian ancestor showed higher inclusion in the brain (Mazin et al, 2021). This work supports the hypothesis that new exons emerge with initial expression in the transcriptionally permissive environment of the testis before co-option into the brain.”

3. Figure 2 suggests the exon evolution events, marked by red triangles, evolved 5 MYA and are human-specific. However, the cited references generally describe them as hominoid-specific (except ref 19). Unless additional evidence is provided, these events could date back ~20 MYA. The figure should be revised to avoid overstating the specificity.

We appreciate the reviewer’s feedback. We have modified the figure to now mark the AS evolution events occurring ~20 MYA. The figure legend has been updated accordingly.

4. Figure 4D: The example of tau isoform regulation in primates (MAPT exon 10) is presented as linked to human phenotype, but the mechanistic connection to a human-specific trait is not fully developed. The authors should clarify how isoform proportion shifts translate to human- or hominoid-specific phenotypes, beyond association with disease states.

We thank the reviewer for this comment. We edited the text to read as:

“Despite these intriguing observations, whether this shift in tau isoform proportion contributed to hominoid-specific phenotypes remains to be determined.”

This is a valuable and timely review that synthesizes a large body of literature, including recent discoveries linking AS to phenotypic evolution. With revisions to strengthen Section I with exon evolution mechanisms, streamline Section II, and clarify figures and examples, the manuscript will provide a stronger conceptual framework and greater impact for readers.

We appreciate the reviewer’s positive assessment and thoughtful suggestions, which have helped us further improve the manuscript.

Dear Dr. Xing,

Thank you for submitting your manuscript for consideration by the EMBO Journal. It has now been seen by the three original referees whose comments are enclosed. As you will see, all the referees express interest in your manuscript and are broadly in favour of publication, pending satisfactory minor revision.

Given the referees' positive recommendations, I would like to invite you to submit a revised version of the manuscript, addressing the remaining comments of all reviewers.

When preparing your letter of response to the referees' comments, please bear in mind that this will form part of the Review Process File, and will therefore be available online to the community. For more details on our Transparent Editorial Process, please review our Editorial Policies page: <https://link.springer.com/partners/embo-press/editorial-policies>

Please note that while we will try to handle the review of your next revision internally in order to expedite the process, as I am not an expert on alternative splicing, I might still reach out to any of the three original referees for advice should any highly specialized issue be raised.

Should you have any questions during the revision process, please do not hesitate to contact me via email.

We generally allow three months as standard revision time. Yet, in light of the relatively minor nature of the new comments and in light of our desire to include your paper in our special issue on molecular ecology and evolution that is currently planned for February, I would deeply appreciate it if you can submit your revised manuscript before the end of the year.

Thank you for the opportunity to consider your work for publication. I look forward to your revision.

Yours sincerely,

Yehu Moran
Academic Editor
The EMBO Journal

Read our guidance for manuscript revisions and related editorial policies: <https://link.springer.com/journal/44318/submission-guidelines#cms-Revised-submissions>

<https://media.springernature.com/original/springer-cms/rest/v1/content/27825798/data/v1>

- a point-by-point response to the referees' comments, with a detailed description of the changes made (as a word file).
- a word file of the manuscript text.
- individual production quality figure files (one file per figure)
- a complete author checklist
- Expanded View files (replacing Supplementary Information)
- a Reagents and Tools Table as part of the Methods section

Please remember: Digital image enhancement is acceptable practice, as long as it accurately represents the original data and conforms to community standards. If a figure has been subjected to significant electronic manipulation, this must be noted in the figure legend or in the 'Methods' section. The editors reserve the right to request original versions of figures and the original images that were used to assemble the figure.

We realize that it is difficult to revise to a specific deadline. In the interest of protecting the conceptual advance provided by the work, we recommend a revision within 3 months (22nd Feb 2026). Please discuss the revision progress ahead of this time with

the editor if you require more time to complete the revisions.

Referee #1:

Congratulations on this very interesting review.

Referee #2:

I thank the authors for their replies to my comments on the first version of the manuscript. I find the paper much improved, but I would like to raise the following additional points:

Page 3: "beak size and shape mirror the selective advantage of"

> Change "of" to "in" (food sources are an environment, not an agent of selection).

Page 4: "These data suggest that lineage-specific changes in AS are predominantly driven by cis-regulatory changes..."

> Please write out the alternative hypothesis here: "as opposed to trans-acting factors in the mouse genome such as RNA-binding proteins" or something similar.

Page 6 First paragraph of section II

> I think it's worthwhile to add a description of the trade-off of using long read sequencing in the first paragraph of section II - that absolute quantification with long reads is still challenging due to limited throughput, so they are more used as a discovery tool, not so much of a quantification tool.

Page 7 and onwards (Section III).

> My comment of an adaptationist view on what are the forces that drive AS patterns and differences therein across species still holds to this section. The authors have replied to my previous comment on this, but made the changes in section IV, which is not what I had recommended. Nearly all differences in AS patterns described here are done so with an adaptationist undertone, implying that there is a selective benefit in e.g. humans to encode for more isoforms than in mouse or vice versa (page 8). Many of the papers cited here are very old and this section overlooks the large, recent literature on neutral evolution of AS diversity. It is in this discussion that I would like to see a more balanced take on what evolutionary forces are responsible for the patterns observed, and at the very least, to state if the cited analyses have even considered the possibility of drift driving these patterns.

Page 10: "This relaxation of purifying selection provides..."

> It would be good to state the alternative interpretation of this (as shown in Benitiere et al. eLife) - if purifying selection is relaxed, as it is in organisms with increasingly small populations, an increase in AS diversity can be interpreted as inefficient selection to remove noisy splicing. The authors are right to note that this is potentially a mechanism of "increasing complexity through drift" as in the case of gene duplicate evolution, but they do not cite the original literature on gene duplications (Force et al 1999 Genetics). A counter-argument has been made that in such cases the selective benefit of a novel isoform should be very large for natural selection to overcome drift and drive the isoform to fixation - I consider the jury to be still out on this. Overall, it would be good to elaborate more on the alternative interpretations.

Referee #3:

The review has been improved. The reviewer has two specific remaining concerns:

1. Section II and Figure 3: These elements remain largely unrelated to the central theme of the review. If the authors wish to retain them, the figure quality should be improved and additional structured information should be provided to clarify their relevance. For example, a table summarizing alternative splicing studies across different technologies (Technology | Studies | Species | Major Findings/Key Features) would help guide readers.
2. Figure 4: The newly added figure should incorporate brain-specific regulatory mechanisms, including downstream gene targets and their contributions to mammalian neural development.

The reviewer believes that addressing these points and improving the overall clarity of presentation will bring the review to a standard suitable for publication in The EMBO Journal.

Detailed Responses to Reviewer Comments

We thank all three reviewers for their thoughtful evaluation of our work. We address each of their additional comments below. All new and revised text is highlighted in red in the revised manuscript.

Referee #1:

Congratulations on this very interesting review.
We thank the reviewer for their positive comment.

Referee #2:

I thank the authors for their replies to my comments on the first version of the manuscript. I find the paper much improved, but I would like to raise the following additional points:

- 1) Page 3: "beak size and shape mirror the selective advantage of"

> Change "of" to "in" (food sources are an environment, not an agent of selection).
We thank the reviewer for this suggestion. The sentence now reads:

“Driven by relative BMP4 expression, beak size and shape mirror the selective advantage in different food sources (Abzhanov *et al*, 2004).”

- 2) Page 4: "These data suggest that lineage-specific changes in AS are predominantly driven by cis-regulatory changes..."

> Please write out the alternative hypothesis here: "as opposed to trans-acting factors in the mouse genome such as RNA-binding proteins" or something similar.
We thank the reviewer for this suggestion. The sentence now reads:

“These data suggest that lineage-specific changes in AS are predominantly driven by cis-sequence changes, rather than by changes in the concentration or activity of trans-acting factors, at least over the evolutionary distance between human and mouse (Barbosa-Morais *et al*, 2012).”

- 3) Page 6 First paragraph of section II

> I think it's worthwhile to add a description of the trade-off of using long read sequencing in the first paragraph of section II - that absolute quantification with long reads is still challenging due to limited throughput, so they are more used as a discovery tool, not so much of a quantification tool.

We thank the reviewer for prompting us to clarify this point. We added a sentence to the first paragraph of Section II to clarify that with continuous improvements in sequencing yield and accuracy, long-read RNA-seq now supports both transcript discovery and quantification. The sentence reads:

“While early long-read RNA-seq studies were constrained by low throughput and high base error rate, continuous improvements in sequencing yield and accuracy have greatly expanded the capabilities of long-read RNA-seq, enabling accurate transcript discovery and quantification (Ament *et al*, 2025).”

4) Page 7 and onwards (Section III).

> My comment of an adaptationist view on what are the forces that drive AS patterns and differences therein across species still holds to this section. The authors have replied to my previous comment on this, but made the changes in section IV, which is not what I had recommended. Nearly all differences in AS patterns described here are done so with an adaptationist undertone, implying that there is a selective benefit in e.g. humans to encode for more isoforms than in mouse or vice versa (page 8). Many of the papers cited here are very old and this section overlooks the large, recent literature on neutral evolution of AS diversity. It is in this discussion that I would like to see a more balanced take on what evolutionary forces are responsible for the patterns observed, and at the very least, to state if the cited analyses have even considered the possibility of drift driving these patterns.

We thank the reviewer for their comment. As noted in our previous revision, our view is that most AS changes over the course of evolution represent stochastic noise and neutral drift of the transcriptome, while only a minority may have potentially acquired adaptive functions. Therefore, we added a paragraph in the previous round that outlines this conceptual framework at the beginning of Section IV, as it provides a natural transition before we highlight a small set of examples linking AS to phenotypic innovation.

In light of the reviewer’s comment, we have made additional revisions to the manuscript to ensure a more balanced and neutral framing. For example, we changed a sentence in the abstract from:

“Large-scale comparative transcriptomic studies have revealed that AS leads to lineage-specific and tissue-specific transcriptomic and proteomic changes, underscoring its adaptive benefits and contribution to organismal complexity.”

to now read:

“Large-scale comparative transcriptomic studies have revealed that AS leads to lineage-specific and tissue-specific transcriptomic and proteomic changes, underscoring its contribution to the evolution of gene products and functions.”

Additionally, we have added a new concluding paragraph to Section III to provide a more balanced view of the evolutionary forces at play, and to note that the high AS complexity observed in large-bodied animals may be explained by random genetic

drift and relaxed purifying selection associated with their small effective population sizes. The paragraph reads:

“A long-standing question in the field is whether AS complexity correlates with and contributes to organismal complexity (Nilsen & Graveley, 2010). Barbosa-Morais et al. found substantial differences in AS complexity among vertebrate species, with the highest levels observed in primates (Barbosa-Morais et al, 2012). Increased AS complexity has also been observed over 1.4 billion years of eukaryotic evolution, with AS complexity strongly correlated with organismal complexity as measured by the number of cell types (Chen et al, 2014). While these observations might invite an adaptationist explanation, it has also been argued that the high AS complexity observed in large-bodied animals can instead be explained by their small effective population sizes, suggesting that random genetic drift and relaxed purifying selection are major driving forces (Bénitière et al, 2024). The relative roles of adaptive evolution versus neutral drift in shaping AS evolution, both at the transcriptome scale and in specific genes, remain to be elucidated.”

5) Page 10: "This relaxation of purifying selection provides..."

> It would be good to state the alternative interpretation of this (as shown in Benitiere et al. eLife) - if purifying selection is relaxed, as it is in organisms with increasingly small populations, an increase in AS diversity can be interpreted as inefficient selection to remove noisy splicing. The authors are right to note that this is potentially a mechanism of "increasing complexity through drift" as in the case of gene duplicate evolution, but they do not cite the original literature on gene duplications (Force et al 1999 Genetics). A counter-argument has been made that in such cases the selective benefit of a novel isoform should be very large for natural selection to overcome drift and drive the isoform to fixation - I consider the jury to be still out on this. Overall, it would be good to elaborate more on the alternative interpretations.

We thank the reviewer for their suggestion. This point is well-taken, and we have added a new concluding paragraph to Section III, referenced in our response above to the previous comment, to provide a more balanced view of the evolutionary forces at play.

Additionally, we have added the Force et al. 1999 citation per the reviewer's suggestion.

Referee #3:

The review has been improved. The reviewer has two specific remaining concerns:

1) Section II and Figure 3: These elements remain largely unrelated to the central theme of the review. If the authors wish to retain them, the figure quality should be

improved and additional structured information should be provided to clarify their relevance. For example, a table summarizing alternative splicing studies across different technologies (Technology | Studies | Species | Major Findings/Key Features) would help guide readers.

We view Section II and Figure 3 as integral components of the review. The technologies described in Section II and illustrated in Figure 3, such as long-read RNA-seq and proteomics, are essential in some of the landmark studies highlighted in Sections III and IV. Figure 3 was intentionally designed to be simple and intuitive, and based on our past experience, we expect it to serve as a useful reference and resource for readers.

- 2) Figure 4: The newly added figure should incorporate brain-specific regulatory mechanisms, including downstream gene targets and their contributions to mammalian neural development. The reviewer believes that addressing these points and improving the overall clarity of presentation will bring the review to a standard suitable for publication in The EMBO Journal.

We thank the reviewer for this comment. To clarify this point, we have revised the manuscript and added an appropriate citation documenting PTBP1's well-established role as a repressor of neural-specific AS events. The revised text now reads:

“For instance, polypyrimidine tract binding protein 1 (PTBP1) is a well-characterized repressor of neural-specific AS events (Keppetipola *et al*, 2012). A mammalian-specific exon skipping event within *PTBP1* was shown to drive AS changes in downstream genes by weakening the functional activity of PTBP1, thereby contributing to mammalian neural development (Figure 4) (Gueroussov *et al*, 2015).”

We would like to note that the study by Gueroussov *et al*. did not functionally characterize any specific PTBP1 target exons and instead focused on the global impact of *PTBP1*'s mammalian-specific exon skipping event. We followed this conceptual framework in Figure 4.

Dear Dr. Xing,

Thank you for submitting your revised manuscript for consideration by the EMBO Journal. I am sorry to bother you with such small things but we would need you to correct the following points:

1. CORRESPONDING AUTHOR: The corresponding author's email address should be added to the line where they are defined on the title page.
2. DISCLOSURE AND COMPETING INTERESTS STATEMENT: please rename according to EMBO Journal standard.
3. FIGURE CALLOUTS: missing for Fig 3A, please add at least one callout for this panel in the main text.

Yours sincerely,

Yehu Moran
Academic Editor
The EMBO Journal

Read our guidance for manuscript revisions and related editorial policies: <https://link.springer.com/journal/44318/submission-guidelines#cms-Revised-submissions>

<https://media.springernature.com/original/springer-cms/rest/v1/content/27825798/data/v1>

- a point-by-point response to the referees' comments, with a detailed description of the changes made (as a word file).
- a word file of the manuscript text.
- individual production quality figure files (one file per figure)
- a complete author checklist

The authors addressed the remaining editorial issues.

Dear Dr. Xing,

I am pleased to inform you that your manuscript has been accepted for publication in the EMBO Journal.

Please note we plan to include your review in our special issue on molecular ecology and evolution that is planned to be published on the second half of March 2026. Yet, it is likely that your paper will become available online much sooner than that.

You may qualify for financial assistance for your publication charges - either via a Springer Nature fully open access agreement or an EMBO initiative. Check your eligibility: <https://link.springer.com/journal/44318/how-to-publish-with-us>

Yours sincerely,

Yehu Moran
Academic Editor
The EMBO Journal

Please note that it is The EMBO Journal policy for the transcript of the editorial process (containing referee reports and your response letters) to be published as an online supplement to each paper. If you should prefer removal of any referee-only figures included in the point-by-point response(s), e.g. because they may still be used for future publication or because they have been reproduced from published work by others, please do let us know immediately via response email.

More information is available here: <https://link.springer.com/partners/embo-press/editorial-policies#Peer%20review>
